# Path Planning for Wheeled Mobile Robot in Partially Known Uneven Terrain

**DOI:** 10.3390/s22145217

**Published:** 2022-07-12

**Authors:** Bo Zhang, Guobin Li, Qixin Zheng, Xiaoshan Bai, Yu Ding, Awais Khan

**Affiliations:** 1College of Mechatronics and Control Engineering, Shenzhen University, Shenzhen 518060, China; zhangbo@szu.edu.cn (B.Z.); 2100292009@email.szu.edu.cn (G.L.); 1810294017@email.szu.edu.cn (Q.Z.); 1910294029@email.szu.edu.cn (Y.D.); awaiskhan@szu.edu.cn (A.K.); 2Shenzhen City Joint Laboratory of Autonomous Unmanned Systems and Intelligent Manipulation, Shenzhen University, Shenzhen 518060, China

**Keywords:** hierarchical path planning, uneven terrain, A^⋆^ algorithm, *Q*-learning algorithm

## Abstract

Path planning for wheeled mobile robots on partially known uneven terrain is an open challenge since robot motions can be strongly influenced by terrain with incomplete environmental information such as locally detected obstacles and impassable terrain areas. This paper proposes a hierarchical path planning approach for a wheeled robot to move in a partially known uneven terrain. We first model the partially known uneven terrain environment respecting the terrain features, including the slope, step, and unevenness. Second, facilitated by the terrain model, we use A${}^{⋆}$ algorithm to plan a global path for the robot based on the partially known map. Finally, the *Q*-learning method is employed for local path planning to avoid locally detected obstacles in close range as well as impassable terrain areas when the robot tracks the global path. The simulation and experimental results show that the designed path planning approach provides satisfying paths that avoid locally detected obstacles and impassable areas in a partially known uneven terrain compared with the classical A${}^{⋆}$ algorithm and the artificial potential field method.

## 1. Introduction

Mobile robots that are deployed for rescue missions in uneven cluttered terrains generally need to have the ability of autonomous navigation and path planning. However, it is challenging to plan a feasible path efficiently for a robot to move in uneven terrains due to the terrains’ slope, step, and unevenness. In [1], a real-time obstacle avoidance method is proposed based on trajectory space, which considers the mobile robot’s uncertainty. However, the uneven terrain modeled in [1] does not reflect a realistic terrain environment well. Some research has been conducted for robotic path planning on uneven terrains, such as the path planning for the Chang’e-4 lunar exploration rover Yutu-2 to move through uneven rough terrain [2]. The Yutu-2 lunar rover can passively adapt to the uneven terrain on the moon’s far side by using its differential mechanism and rocker arm. This configuration enables the rover to reduce its pitch angle by half compared with other vehicles when clearing an obstacle. The Curiosity rover used in the US Mars exploration mission is a six-wheeled vehicle [3]. It uses a rocker arm steering structure in which two front wheels and two rear wheels are independent such that the rover can pivot steering. The Curiosity rover, meanwhile, relies on a six-wheeled primary and secondary joystick system to navigate over the uneven rocks of Mars. The above work focuses on vehicles’ adaptation and safe travel over rough terrain through sophisticated mechanical mechanisms. However, it must be acknowledged that this is a complex and costly approach. The application of path planning techniques would be much more beneficial if they could be used to continually identify whether they can be safely navigated to plan safe and feasible paths in rough environments.

For the path planning of wheeled mobile robots in uneven terrains, a high-quality terrain model is the foundation for path planning as the fidelity of the model affects the applicability of the planned path [4]. A lot of work has been performed on modeling uneven terrain environments. In [5], Dupuis et al. constructed a three-dimensional (3D) terrain by using an ILRIS-3D laser scanner to measure the unevenness of the road surface. Vandapel et al. used a 3D laser radar to obtain the environmental data of uneven terrain, and classified the uneven terrain by learning rocks’ characteristics [6]. Agrawal et al. equipped a stereo vision camera on a wheeled mobile robot to map unknown uneven terrain [7]. The wheeled mobile robot moves to a target position in a new environment, where real-time correction is made in the map construction process. Huber et al. modeled uneven terrain by collecting data from three parts: the internal environment map constructed from the depth camera data, the large-scale topographic map constructed from the aerial data, and the map built from the stereo data [8]. The above research work shows that the grid-based 3D elevation map can be used to model an uneven terrain.

The goal of the path planning for a robot operating in a challenging uneven terrain is to find the optimal feasible path that avoids impassable areas, including obstacles, and reduces failure risk due to the terrain’s slope, step, and unevenness. Different kinds of algorithms have been designed for robot path planning, including the Dijkstra algorithm [9], A${}^{⋆}$ algorithm [10], RRT algorithm [11], artificial potential field (APF) based algorithms [12], and intelligent algorithms such as genetic algorithm [13], ant colony algorithm [14], particle swarm optimization algorithm [15], and reinforcement learning based algorithms [16]. Pan and Xu used an improved A${}^{⋆}$ algorithm for path planning in a 3D terrain, where a penalty function is used to guarantee the robot’s stability [17]. Peng et al. applied an improved APF method in a 3D space and introduced a tangent point for obstacle avoidance [18]. The repulsive force generated by the tangential point can guide a robot to its goal position and reduce the algorithm’s running time. Pan et al. proposed a hybrid genetic ant algorithm to improve the efficiency of path planning in a 3D terrain [19]. Josef and Degani proposed a deep reinforcement learning method for local path planning of a robot in unknown uneven terrain in [20]. In [21], a map-based offline path planning approach was designed to construct an initial path for quadrotor UAVs, followed by a vision-based obstacle detection method using optical flow for collision avoidance. Online and real-time processing experiments demonstrate the feasibility of the proposed method for path planning and autonomous navigation. A hybrid navigation system was developed in [22] for a two-wheel differential drive mobile robot that includes static-environment global path planning and dynamic-environment obstacle-avoidance tasks, where a multi-agent A-heuristic algorithm was proposed for finding the optimal obstacle-free path and a weighted-sum model was employed to adapt to the dynamic obstacles.

This paper studies the path planning for a wheeled robot that operates in a partially known uneven terrain with incomplete environmental information such as locally detected obstacles and impassable terrain areas. First, the A${}^{⋆}$ algorithm is used for the robot’s global path planning based on partially known environmental information. Then, an adaptable *Q*-learning method is designed for the robot’s local path planning to avoid locally detected obstacles and impassable terrain areas when the robot tracks the global path. The main contributions of this paper are as follows. First, we modeled the partially known uneven terrain considering its slope, step, and unevenness. Second, to avoid the robot colliding with locally detected obstacles and entering impassable terrain areas, we proposed a hierarchical path planning approach by integrating the A${}^{⋆}$ algorithm for the robot’s global path planning with the *Q*-learning algorithm for the robot’s local path planning. The proposed approach has satisfying performance compared with the classical A${}^{⋆}$ algorithm and the artificial potential field method.

The rest of the paper is organized as follows. In Section 2, we model the uneven terrain environment and formulate the path planning problem. The A${}^{⋆}$ algorithm is presented as the global path planning algorithm, and the *Q*-learning algorithm is introduced for the local path planning in Section 3. Section 4 shows the numerical and experimental results of the proposed path planning approach. Section 5 concludes the paper.

## 2. Environmental Modeling and Problem Formulation

In this section, the 3D digital elevation model (DEM) is first employed to describe the detailed geological surface information of the uneven terrain as in [23]. The 3D DEM can digitally represent the surface information of uneven terrains by considering the terrains’ elevation data. The terrain elevation data can be collected from different types of sensors, such as LIDAR, depth camera, ultrasonic, and terahertz [24]. Second, facilitated by the environmental modeling, we formulate the path planning problem and present the kinematic model of the wheeled robot.

### 2.1. Digital Representation of 3D DEM

The 3D DEM is a raster data model that stores the elevation information of a set of terrain positions in digital form to digitally express the morphology of uneven terrain surfaces. It describes the spatial distribution of an uneven terrain environment by using a set of 3D vectors {x,y,z}, where *x* and *y* represent the plane position of mapping an environmental feature point of the uneven terrain, and *z* is the height of the point. Therefore, geomorphic characteristics such as the slope aspect, slope change rate, step, and unevenness can be well extracted by 3D DEM [25].

### 2.2. Information Extraction of a Partially Known Uneven Terrain Environment

This section models the partially known uneven terrain environment considering its geomorphic features such as slope, step, and unevenness. Slope describes the inclining degree of the uneven terrain, where a greater slope value implies a more inclining uneven terrain that is more difficult for a wheeled mobile robot to pass. Step indicates the change rate of the vertical height of the uneven terrain within a certain horizontal distance. The larger the step value is, the more likely the wheeled mobile robot will overturn. The unevenness describes the jitter degree of the terrain’s elevation, where a greater unevenness value would lead to a more unstable motion of the wheeled mobile robot.

The terrain’s slope actually represents the angle between the terrain plane and the horizontal plane, which can be calculated by the angle between the vertical *Z* axis and the normal vector of the terrain plane. Assume that $N^{2}$ points are sampled, where the sampling point located at the *i*-th row and *j*-th column of the raster map is $\left(x_{i,j},y_{i,j},z_{i,j}\right)$, i,j∈S, S={1,2,…,N}, and $z_{i,j}$ is the point’s elevation value. Then, the equation of the terrain plane can be defined as
(1)Ax+By+C=z,
where *A*, *B*, and *C* are the coefficients to be solved. Meanwhile, the least squares equation is obtained as
(2)$$F=\sum_{i,j\in S}{\left(Ax_{i,j}+By_{i,j}+C-z_{i,j}\right)}^{2}.$$

The partial derivatives of *F* with respect to *A*, *B*, and *C* are respectively zero, where the equation of the terrain plane can be achieved after calculating *A*, *B*, and *C* using the least square regression. The normal vector of the terrain plane is $\vec{n}=\left(A,B,-1\right)$, whereas for the horizontal plane it is $\vec{m}=\left(0,0,1\right)$. The angle between the normal vector and horizontal plane of grid *i* is defined as φ, satisfying the angle formula between a line and plane as follows
(3)$$\begin{array}{cc} \sin\varphi_{i} & =\frac{\vec{n}\cdot \vec{m}}{\left|\vec{n}\right|\cdot \left|\vec{m}\right|}. \end{array}$$

The slope angle of the grid *i* of the terrain plane is
(4)$$\theta_{i}=\frac{\pi }{2}-sin^{-1}\frac{\vec{n}\cdot \vec{m}}{\left|\vec{n}\right|\cdot \left|\vec{m}\right|}.$$

The unevenness of the uneven terrain dramatically impacts the state stability of the wheeled robot as the robot might roll over if the terrain’s unevenness is too large. In this paper, the mean-square deviation of the road elevation is used to quantify the unevenness information considering the influence of the wheel diameter of the wheeled robot. Then, the unevenness value of the current grid/point *i* is
(5)$$\omega_{i}=D^{-1}\sqrt{\frac{1}{9}\sum_{J\in U_{i}}{\left(z\left(J\right)-\bar{z}\right)}^{2}},$$
where *D* is the wheel diameter of the wheeled mobile robot, z(J) is the elevation value corresponding to a plane position *J* in set $U_{i}$, and $\bar{z}$ is the mean elevation value of all the plane positions in $U_{i}$. $U_{i}$ contains the plane position information of the current grid *i* and its eight neighbor grids.

### 2.3. Objective Function

The traversability of the *i*-th grid of the map is indicated as $T_{i}\in \left[0,1\right]$, which represents the easiness for the robot to pass considering the grid’s slope, step, and unevenness as
(6)$$T_{i}=k_{1}\cdot \frac{\theta_{i}}{\theta_{crit}}+k_{2}\cdot \frac{\delta_{i}}{\delta_{crit}}+k_{3}\cdot \frac{\omega_{i}}{\omega_{crit}}.$$

The sum of the three non-negative parameters $k_{1}$, $k_{2}$, $k_{3}$ is 1, and $\theta_{crit},\delta_{crit},\omega_{crit}$ respectively represent the maximum unevenness characteristic factors that the robot can pass through each grid. In particular, the step value $\delta_{i}$ is the maximum z-direction elevation difference between the *i*-th grid and its adjacent eight grids. When any grid *i*’s slope value $\theta_{i}$, step value $\delta_{i}$, or unevenness value $\omega_{i}$ reaches its maximum, set $T_{i}=1$, implying that the grid is impassable.

Let the traversability cost for traveling between the two grids *i* and *j* be
(7)$$T_{i,j}=\sqrt{{\left(T_{i}-T_{j}\right)}^{2}}.$$

Let the Euclidean distance between two grids *i* and *j* be $l_{i,j}$. Then, the objective function of the path planning problem considers both the distance and traversability cost of the planned path as
(8)$$f=\sum_{k=1}^{M-1}\left(l_{k,k+1}+w\cdot T_{k,k+1}\right).$$
where *w* is the parameter used to adjust the weight of the traveled distance and the road traversability, *k* is the *k*-th grid on the planned path, and *M* is the total number of grids traversed by the planned path. The second term of Equation (Equation 8) considers the traversability cost of the planned path, which can enable the robot to escape impassable areas. The travel cost *f* is positive-infinitely large if any grid *i* of the planned path satisfies $T_{i}=1$.

### 2.4. Robot Kinematic Model

As [26], we assumed that the kinematics of the four-wheel mobile robot is
(9)$$\left[\begin{array}{c} \dot{x}\\ \dot{y}\\ {\dot{\theta }}_{a} \end{array}\right]=\left[\begin{array}{cc} \cos\left(\theta_{a}\right) & 0\\ \sin\left(\theta_{a}\right) & 0\\ 0 & 1 \end{array}\right]\left[\begin{array}{c} v\\ m \end{array}\right],$$
where *v*, *m*, *x*, *y*, and $\theta_{a}\in \left[-\pi ,\pi \right]$ respectively represent the robot’s tangential velocity, angular velocity, horizontal displacement, vertical displacement, and yaw angle. In the simplified differential car model, the right and left wheel velocities are defined as $v_{R}$ and $v_{L}$. Then, *v* and *m* in Equation (Equation 9) can be calculated as
(10)$$v=\frac{v_{L}+v_{R}}{2},\;\;m=\frac{v_{R}-v_{L}}{\lambda },$$
where λ is the distance between the two wheels and $v\leq v_{\max},\;\;\left|m\right|\leq \frac{2v_{\max}}{\lambda }$. $v_{max}$ is the robot’s maximum speed.

## 3. Path Planning Algorithm for Mobile Robot

Path planning for autonomous mobile robots in a partially known environment generally contains two procedures: global path planning and local path planning. Global path planning is a prior planning for a global path for the robot to move from a start position to a specific goal position based on prior known environmental information. The applicability of the planned global path depends on the accuracy of the environmental information. If some initially unknown obstacles and impassable terrain areas are detected by the robot’s on-board sensors when tracking the global path, local path planning is needed to dynamically adjust the robot’s path.

### 3.1. A${}^{⋆}$ Algorithm for Global Path Planning

The classical A${}^{⋆}$ algorithm is a graph search algorithm typically used for global path planning [27]. It can achieve the shortest path for a robot to move between two prescribed positions in a known static environment.

To enable the robot to automatically avoid impassable areas due to slope, step and unevenness, in this paper the cost function of A${}^{⋆}$ considering the traversability cost is defined as
(11)$$F\left(n\right)=G_{s}\left(n\right)+H_{g}\left(n\right),$$
where *n* is the index of the current grid reached by the planned path, $G_{s}\left(n\right)$ is the minimum accumulated cost for the robot to move from its start grid *s* to the current grid *n*, and $H_{g}\left(n\right)$ is the estimated minimum cost for the robot to move from *n* to the goal grid *g*.

The cost function F(n) is applied to the A${}^{⋆}$ algorithm for the global path planning in a 3D uneven terrain. According Equation (Equation 8), the accumulated travel cost $G_{s}\left(n\right)$ considering the influence of the terrain’s slope, step, and unevenness is
(12)$$G_{s}\left(n\right)=L_{sn}+w\cdot T_{sn},$$
where $L_{sn}=\sum_{k=1}^{n-1}l_{k,k+1}$ is the accumulated Euclidean distance for the robot to travel from the start grid *s* to the current grid *n*, and $T_{sn}=\sum_{k=1}^{n-1}T_{k,k+1}$ is the accumulated traversability cost for the robot to travel between the two grids.

To guarantee that the A${}^{⋆}$ algorithm is optimal, the heuristic function $H_{g}\left(n\right)$ can be set as the Euclidean distance $l_{n,g}$ between *n* and *g*.

### 3.2. Q-Learning Based Local Path Planning Algorithm

In a known static environment, the wheeled mobile robot can safely track the initially planned global path resulting from the A${}^{⋆}$ algorithm. However, in a partially known uneven environment, the robot might encounter locally detected obstacles and impassable terrain areas that can only be detected within the sense distance of the robot. If the wheeled mobile robot cannot avoid these locally detected obstacles and impassable terrain areas well, it might fail to reach its destination. In [28], a local path planning method was designed based on a *Q*-learning algorithm for a robot to avoid locally detected obstacles.

According to [29], the *Q*-learning algorithm can optimize the local path of a wheeled mobile robot whenever an obstacle or impassable area located on the global planned path is detected through the robot’s interactive exploration and evaluation. The locally planned path resulting from the collision avoidance maneuver is latched to the nearest state on the already computed global path. A higher feedback reward of performing an action in the environment implies a better action: the action will be performed more in the future, and vice versa. The feedback of the unknown uneven terrain environment is obtained through constant trial and error. Inspired by [30], we design the schematic diagram of the interaction between the robot and the environment in Figure 1.

*Q*-learning algorithm is a value-based reinforcement learning algorithm, where *Q* denotes the corresponding action value the robot can obtain by taking a specified action at a certain moment. Inspired by [31], we construct the *Q*-learning algorithm in Algorithm 1, which uses a matrix Q to store the *Q* values resulting from the robot’s action $a_{t}$ at each state/grid. The robot’s action space at each grid contains eight moving directions: up, down, left, right, upper-left, upper-right, lower-right, and lower-left. The matrix Q is employed to evaluate the corresponding actions to be taken in each current state: if the *Q* value for performing an action at a state is higher, it is better to take action at this state, and vice versa. For each time step *t* as shown in Algorithm 1, the robot chooses an action $a_{t}\in A$ under its current state $s_{t}\in S$ based on the ε-greedy strategy. The strategy maps the relationship between state $s_{t}$ and action $a_{t}$. Meanwhile, the robot obtains a reward $r_{t}$ according to the reward function and evolves to the next state $s_{t+1}$ based on the current state $s_{t}$ and action $a_{t}$. In an episode of Algorithm 1, the robot continues the cyclic process shown in Figure 1 until one of the robot’s following situations happens: (1) getting to the goal position; (2) colliding with an obstacle; and (3) moving into an impassable grid. When an obstacle is detected, the reward for the robot entering the obstacle area is set to −1, implying a punishment is incurred if the robot encounters the obstacle.
**Algorithm 1 ***Q*-learning algorithm**Initialize**$Q_{t}\left(s_{t},a_{t}\right),\forall s_{t}\in S,a\in A\left(s\right),maximum\;training\;episode\;number\;E,Q($terminalstate, ·
)=0
1:**for** every episode value of {1,2,...,E} **do**2:    Initialize state $s_{t=0}$3:    **while** the robot does not reach the goal position and does not collide with an obstacle and does not move into an impassable grid **do**4:        Use ε-greedy policy to select an action $a_{t}$5:        obtain the corresponding reward $r_{t}$ and new state $s_{t+1}$ based on the current state $s_{t}$ and action $a_{t}$6:        $Q_{t}\left(s_{t},a_{t}\right)=\left(1-\alpha \right)Q_{t}\left(s_{t},a_{t}\right)+\alpha \left(r_{t}+\gamma \max_{a\in A}Q_{t+1}\left(s_{t+1},a\right)\right)$7:        t←t+18:    **end while**9:**end for**

After the training by using *Q*-learning algorithm, the *Q*-value matrix Q can be obtained. Then, the robot’s action with the maximum *Q* value at each current state can be chosen until reaching the goal position. Based on the *Q*-learning algorithm designed in [32], when the robot performs an action $a_{t}$ in state $s_{t}$, the corresponding action value function $Q_{t}\left(s_{t},a_{t}\right)$ can be updated as
(13)$$Q_{t}\left(s_{t},a_{t}\right)=\left(1-\alpha \right)Q_{t}\left(s_{t},a_{t}\right)+\alpha \left(r_{t}+\gamma \max_{a\in A}Q_{t+1}\left(s_{t+1},a\right)\right),$$
where γ∈[0,1] is the attenuation rate representing the attenuation of future rewards, and α∈(0,1) is the learning rate. The attenuation rate affects the ratio that the robot replaces the original *Q* value with a new value. In addition, the reward function $r_{t}$ in Equation (Equation 13) is defined as
(14)$$r_{t}=\left\{\begin{array}{cc} v_{\mathrm{reward}}, & \;\mathrm{if}\;\mathrm{robot}\;\mathrm{gets}\;\mathrm{to}\;\mathrm{the}\;\mathrm{goal}\;\mathrm{position},\;\\ v_{\mathrm{penalty}}, & \;\mathrm{if}\;\mathrm{robot}\;\mathrm{collides}\;\mathrm{to}\;\mathrm{an}\;\mathrm{obstacle},\;\;\\ -G_{s}\left(n\right), & \;\mathrm{others}.\; \end{array}\right.$$
where $v_{\mathrm{reward}}$ is a positive value indicating a reward, and $v_{\mathrm{penalty}}$ is a negative value indicating a penalty. When the robot cruises on an uneven terrain, the reward $G_{s}\left(n\right)$ is calculated according to Equation (Equation 12).

The term $\max_{a\in A}Q_{t+1}\left(s_{t+1},a\right)$ is the optimal action value of $Q_{t+1}\left(s_{t+1},a\right)$ at the next time step corresponding to all possible actions *a*, which is denoted as
(15)$$Q_{t+1}^{*}\left(s_{t+1},a_{t+1}\right)=\max_{a\in A}\left(Q_{t+1}\left(s_{t+1},a\right)\right).$$
$Q_{t+1}^{*}\left(s_{t+1},a_{t+1}\right)$ can be assumed to remain constant for future determined state $s_{t+1}$ corresponding to the optimal action $a_{t+1}$.

Through a certain number of episodes of learning and training as shown in Algorithm 1, the robot learns new knowledge by constantly interacting with the environment until the convergence of the *Q* values $Q_{t}^{*}\left(s_{t},a_{t}\right)$. Based on Equation (Equation 13), we can obtain
(16)$$\begin{array}{cc} Q_{t}^{*}\left(s_{t},a_{t}\right)= & {\left(1-\alpha \right)}^{n}Q_{t}\left(s_{t},a_{t}\right)+{\left(1-\alpha \right)}^{n-1}\alpha \left(r_{t}+\gamma Q_{t+1}^{*}\left(s_{t+1},a_{t+1}\right)\right)\\ \; & +\ldots +\alpha \left(r_{t}+\gamma Q_{t+1}^{*}\left(s_{t+1},a_{t+1}\right)\right). \end{array}$$

Then, it is straightforward that
(17)$$\begin{array}{cc} Q_{t}^{*}\left(s_{t},a_{t}\right)= & {\left(1-\alpha \right)}^{n}Q_{t}\left(s_{t},a_{t}\right)+\alpha \left(r_{t}+\gamma Q_{t+1}^{*}\left(s_{t+1},a_{t+1}\right)\right)\sum_{i=0}^{n-1}{\left(1-\alpha \right)}^{i}. \end{array}$$

When *n* is an infinite large number, ${\left(1-\alpha \right)}^{n}$ approaches to zero, and $\sum_{i=0}^{n-1}{\left(1-\alpha \right)}^{i}$ approaches to 1/α as (1−α)∈(0,1). Then, equation Equation (Equation 17) leads to
(18)$$Q_{t}^{*}\left(s_{t},a_{t}\right)=r_{t}+\gamma Q_{t+1}^{*}\left(s_{t+1},a_{t+1}\right).$$

According to Equation (Equation 13), we have
(19)$$Q_{t}\left(s_{t},a_{t}\right)=Q_{t}\left(s_{t},a_{t}\right)+\alpha \left(r_{t}+\gamma Q_{t+1}^{*}\left(s_{t+1},a_{t+1}\right)-Q_{t}\left(s_{t},a_{t}\right)\right).$$

Combining Equations (Equation 18) and (Equation 19), when the learning process converges, we obtain
(20)$$Q_{t}\left(s_{t},a_{t}\right)=Q_{t}^{*}\left(s_{t},a_{t}\right).$$

## 4. Simulation and Experimental Tests

We evaluate the performance of the proposed path planning algorithms through both simulation and experimental tests. First, the simulation tests are performed on an Intel(R) Core (TM) i7-6700HQ CPU 2.60 GHz with 8.00 GB RAM, and the algorithms are compiled and implemented by MATLAB under Windows 10. To distinguish from the classical A${}^{⋆}$ which only focuses on minimizing the length of the planned path, the A${}^{⋆}$ algorithm considering the traversability cost of the planned path is simplified as an improved A${}^{⋆}$ algorithm. The improved A${}^{⋆}$ algorithm is first used to plan a global path for the robot moving in an uneven terrain based on partially known terrain information. The global path acts as a reference path for the robot to track. During the path-tracking process, the *Q*-learning algorithm is used to avoid locally detected obstacles and impassable terrain areas. The *Q*-learning algorithm is triggered whenever an obstacle or impassable area on the global planned path is detected. The maximum unevenness characteristic factors that the robot can pass through each grid are respectively $\theta_{crit}=30^{\circ }$, $\delta_{crit}=0.04$ m, and $\omega_{crit}=0.025$ m according to the robot’s physical structure. The range of the whole terrain map is 11 m ×11 m, where the size of the single grid is 0.1 m ×0.1 m. In subsequent experiments, the $v_{\mathrm{reward}}$ and $v_{\mathrm{penalty}}$ in Equation (Equation 14) are set to 10 and −1, respectively.

### 4.1. Global Path Planning Based on A${}^{⋆}$ Algorithm

We first test the performance of the classical A${}^{⋆}$ and the improved A${}^{⋆}$ for path planning in 10 different scenarios of an initially known uneven terrain, where the start position and goal position of the global path planning are randomly generated in each scenario. The improved A${}^{⋆}$ algorithm is performed under different weights *w* of Equation (Equation 8). In Equation (Equation 6), we set $k_{1}=0.2$, $k_{2}=0.4$, and $k_{3}=0.4$ since the step and unevenness generally play a more important role than slope.

Table 1 shows the average performance of the two algorithms, where the improved A${}^{⋆}$ algorithm achieves feasible paths in all 10 scenarios while the classical A${}^{⋆}$ algorithm fails to plan a feasible path in 7 out of 10 scenarios. The average path cost of the 3 feasible paths resulting from the classical A${}^{⋆}$ algorithm is the sum of the initially minimized path length and the traversability cost of the planned path, which is larger than those resulting from the improved A${}^{⋆}$ algorithm when w=1. Furthermore, the travel cost of the planned path resulting from the improved A${}^{⋆}$ algorithm increases with the increase in the weight *w*. This might be because a larger weight *w* mainly generates a smother path while neglecting the planned path as shown in Figure 2, which is consistent with Equation (Equation 8). The traversability information of the uneven terrain is reflected by the color bar on the right part of Figure 2: a brighter color implies that it is more difficult for robot to travel on the uneven terrain, and a grid is unfeasible for the robot to move through if its associated color corresponds with value 1.

In Figure 2, the red-colored path results from the classical A${}^{⋆}$ algorithm, which has the shortest length. However, the path is not feasible as it passes through the yellow area (impassable area), where the robot will roll over if tracking the planned global path. The green-colored path, resulting from the improved A${}^{⋆}$ algorithm under w=5, has a relatively smoother path through the uneven terrain compared with the case when w=1. The blue-colored path, resulting from the improved A${}^{⋆}$ algorithm under w=1, has a shorter path length than the case when w=5. However, the path passes over more terrain areas with a higher traversability cost, implying a higher chance of rollover if the robot tracks the planned global path.

### 4.2. Local Path Planning Based on *Q*-Learning Algorithm

For the local path planning, we test the algorithms’ performance under four different scenarios compared with the artificial potential field method (APF) adapted from [33], the classical A${}^{⋆}$ algorithm, and the improved A${}^{⋆}$ algorithm proposed for global path planning. The start position, the goal position of the global path planning, and the position of an initially unknown obstacle are randomly generated in four scenarios. In Figure 3, the black-colored path is the global path initially planned by the improved A${}^{⋆}$ algorithm with w=1 based on the partially known environmental information in scenario 1. In Figure 3, an initially unknown yellow-colored cylindrical obstacle, locating on the global path, can be locally detected when the robot tracks the global path. The obstacle has a radius of 1 m and a height of 3 m. We assume that the sensing radius of the robot is 0.5 m, and the robot is driven by a constant speed of 1 m/s. It is straightforward to check that the wheeled mobile robot would collide with the locally detected obstacle if just tracking the global path.

Table 2 shows that the total path cost *f* in objective function Equation (Equation 8) corresponding to the local path resulting from the *Q*-learning algorithm in each scenario is the lowest compared with the APF, classical A${}^{⋆}$ algorithm, and improved A${}^{⋆}$ algorithm under different *w*. However, the payoff is the algorithm’s longer running time. Table 2 also shows that the total path cost of the planned path resulting from the improved A${}^{⋆}$ algorithm increases when increasing the weight of *w*, which is consistent with Table 1. Figure 4 and Table 2 show that the locally planned path resulting from the *Q*-learning algorithm has well considered the terrain’s unevenness when avoiding the locally detected obstacle.

### 4.3. Experimental Test

This section experimentally tests the performance of the designed hierarchical path planning approach for guiding a wheeled mobile robot to move in a partially known uneven terrain. Figure 5 shows the Lidar-built DEM of the uneven terrain, where the performance of the algorithms is tested on two experimental scenarios Exp 1 and Exp 2, as shown in Table 3. In the two scenarios, one initially unknown obstacle is locally detected in Exp 1 as shown in Figure 6, whereas two initially unknown obstacles are consecutively detected in Exp 2 as shown in Figure 7.

Figure 6 and Figure 7 respectively show the robot’s path guided by the integrated improved A${}^{⋆}$ algorithm and the *Q*-learning algorithm. The two Appendix A show how the robot dynamically adjusts the initially planned global path to avoid locally detected obstacles in Exp 1 and Exp 2. It can be seen in the videos that the robot can move smoothly from a given start position to a goal position while dynamically adjusting its path from the initially planned global path to avoid locally detected obstacles.

Figure 8 and Figure 9 respectively show the globally planned paths resulting from the classical A${}^{⋆}$ algorithm, the improved A${}^{⋆}$ algorithm, and the robot’s actual path resulting from the integrated improved A${}^{⋆}$ algorithm and the *Q*-learning algorithm, which are complied in MATLAB on experimental scenarios Exp 1 and Exp 2. Figure 8 and Figure 9 show that the wheeled robot can successfully escape the locally detected obstacles when tracking the planned global path, which is consistent with Figure 6 and Figure 7, respectively. The above simulation and experimental results demonstrate that the integrated A${}^{⋆}$ algorithm and the *Q*-learning algorithm can enable the wheeled mobile robot to move smoothly in a partially known uneven terrain while dynamically adjusting its path to avoid locally detected obstacles and impassable terrain areas.

## 5. Conclusions

This paper investigates the path planning of a wheeled mobile robot in a partially known uneven terrain. Based on the initial partially known environmental information, an improved A${}^{⋆}$ algorithm is first used to achieve a global path for the robot to track considering the terrain’s slope, step, and unevenness. Then, the *Q*-learning method is applied to adjust the robot’s local path dynamically to avoid locally detected obstacles and impassable terrain areas when the robot tracks the global path. The simulation and experimental results show the satisfying performance of the integrated improved A${}^{⋆}$ algorithm and *Q*-learning method for guiding the robot’s movement in a partially known uneven terrain compared with the classical A${}^{⋆}$ algorithm and the artificial potential field method. Future work will focus on improving the *Q*-learning method’s efficiency for online obstacle-avoidance path planning.

## Figures and Tables

**Figure 1 sensors-22-05217-f001:**
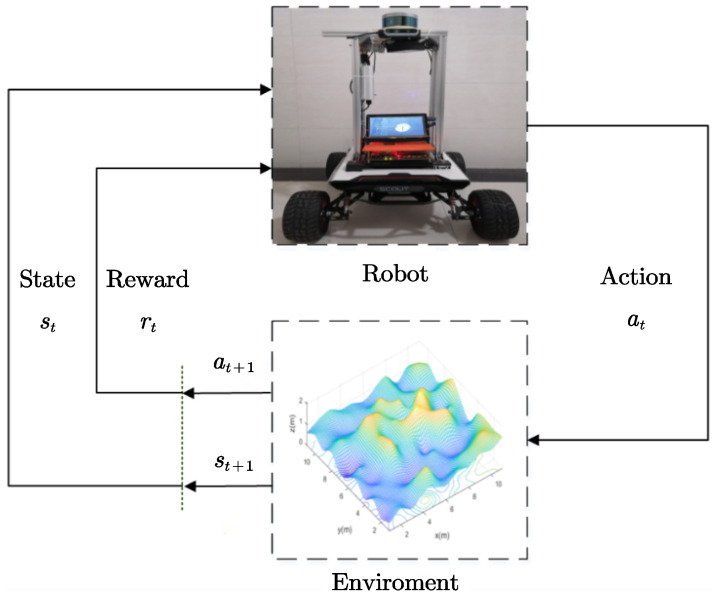
Schematic diagram of interaction between the robot and environment in reinforcement learning.

**Figure 2 sensors-22-05217-f002:**
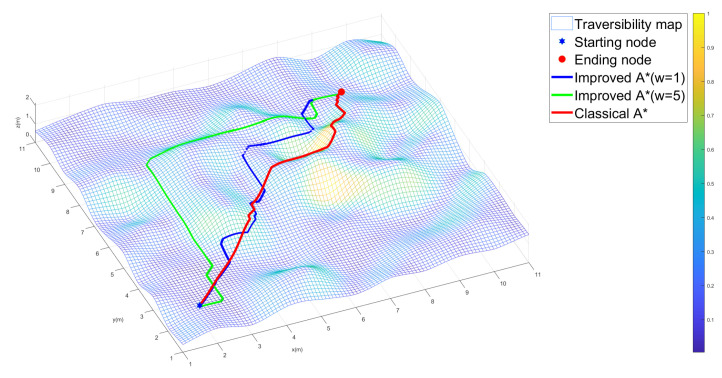
Global path planned by each algorithm in one scenario.

**Figure 3 sensors-22-05217-f003:**
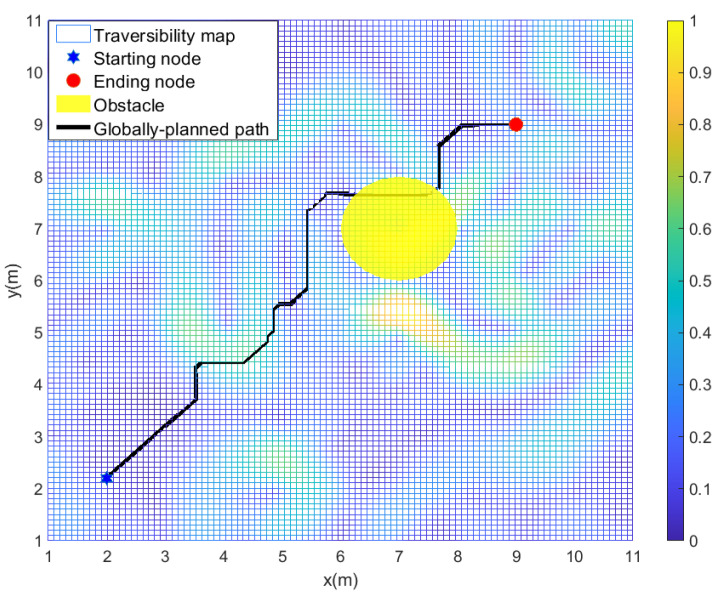
The globally planned path and locally detected obstacle in scenario 1.

**Figure 4 sensors-22-05217-f004:**
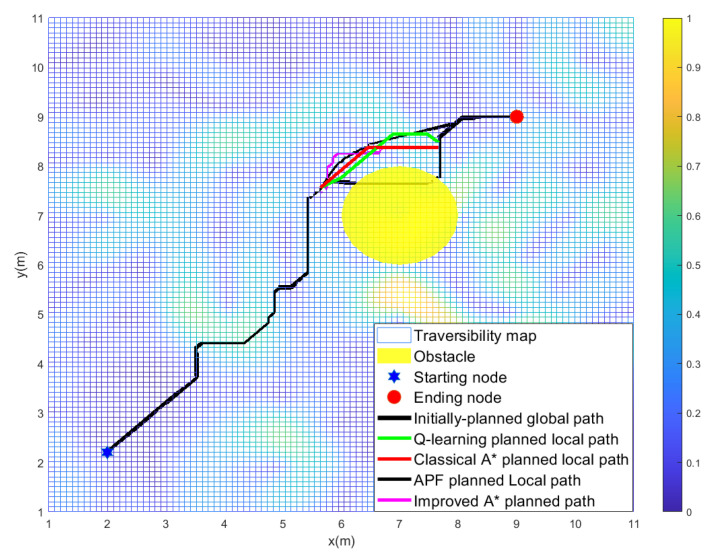
The locally planned path resulting from each algorithm in scenario 1.

**Figure 5 sensors-22-05217-f005:**
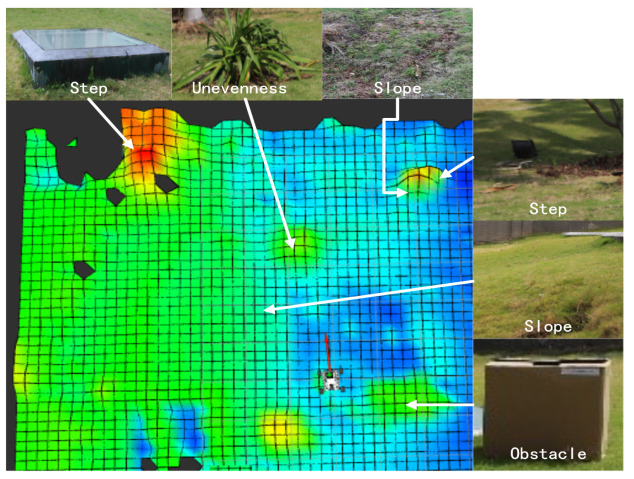
DEM of the experimental uneven terrain.

**Figure 6 sensors-22-05217-f006:**
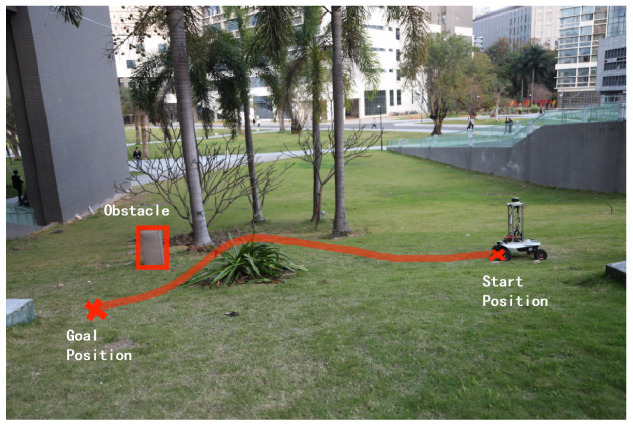
The robot’s path in Exp 1 resulting from the integrated improved A${}^{⋆}$ algorithm and *Q*-learning algorithm.

**Figure 7 sensors-22-05217-f007:**
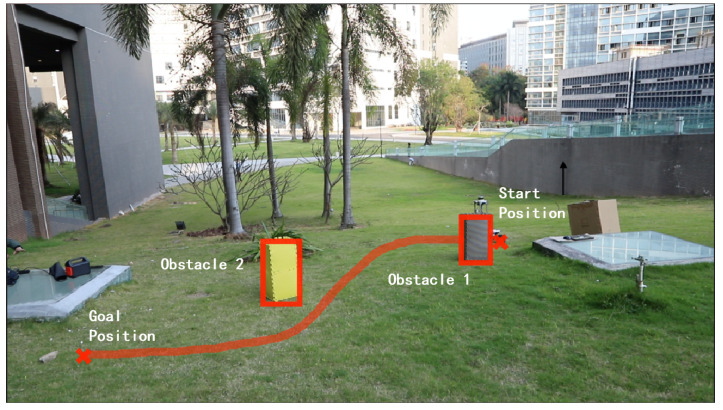
The robot’s path in Exp 2 resulting from the integrated improved A${}^{⋆}$ algorithm and *Q*-learning algorithm.

**Figure 8 sensors-22-05217-f008:**
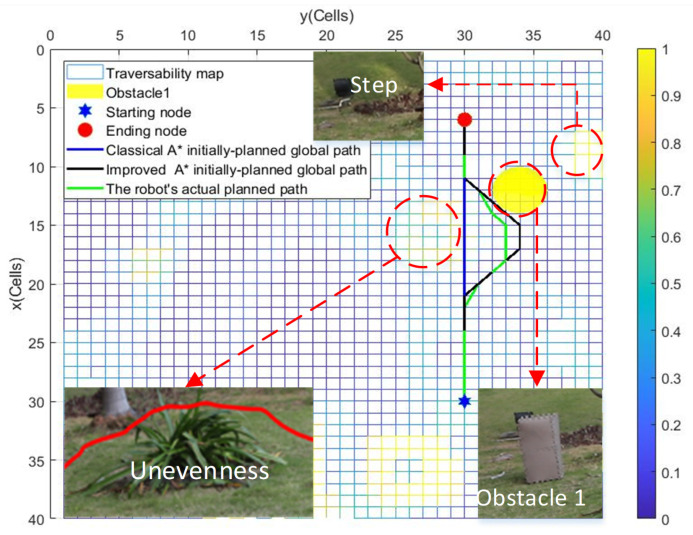
The planned paths in Exp 1.

**Figure 9 sensors-22-05217-f009:**
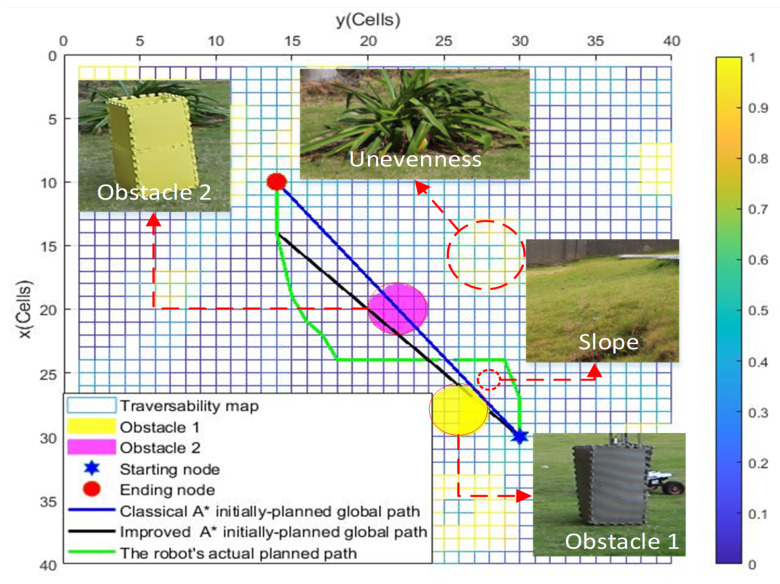
The planned paths in Exp 2.

**Table 1 sensors-22-05217-t001:** The algorithms’ average performance on global path planning in the 10 test scenarios.

Algorithm	Weight	Running	Path	Feasible
	*w*	Time (s)	Cost *f*	Times
Classical A${}^{⋆}$	0	1.42	34.33	3
Improved A${}^{⋆}$	0.2	1.31	28.12	10
	0.5	1.73	31.24	10
	1	1.81	32.17	10
	2	1.89	35.27	10
	4	1.96	38.24	10
	5	2.15	39.51	10

**Table 2 sensors-22-05217-t002:** Overall performance of the algorithms.

Scenarios	Local Path	Weight	Running	Path
	Planning Algorithm	*w*	Time (s)	Cost *f*
Scenario 1	*Q*-learning	1	3.06	29.95
	APF	-	1.45	34.76
	Classical A${}^{⋆}$	0	0.26	37.98
	Improved A${}^{⋆}$	0.2	0.22	31.23
		0.5	0.24	31.78
		1	0.28	32.15
		2	0.33	33.45
		4	0.45	34.25
		5	0.58	35.12
Scenario 2	*Q*-learning	1	2.31	20.36
	APF	-	1.25	28.67
	Classical A${}^{⋆}$	0	0.23	32.82
	Improved A${}^{⋆}$	0.2	0.16	23.51
		0.5	0.16	24.18
		1	0.17	24.97
		2	0.19	25.82
		4	0.21	26.48
		5	0.25	27.69
Scenario 3	*Q*-learning	1	3.21	35.25
	APF	-	1.35	42.32
	Classical A${}^{⋆}$	0	0.31	43.58
	Improved A${}^{⋆}$	0.2	0.26	39.51
		0.5	0.26	40.24
		1	0.28	40.95
		2	0.31	41.46
		4	0.42	43.65
		5	0.49	44.28
Scenario 4	*Q*-learning	1	2.75	23.59
	APF	-	1.14	27.18
	Classical A${}^{⋆}$	0	0.28	31.57
	Improved A${}^{⋆}$	0.2	0.19	27.96
		0.5	0.20	28.53
		1	0.22	29.37
		2	0.24	30.92
		4	0.28	32.51
		5	0.31	33.24

**Table 3 sensors-22-05217-t003:** Two experimental scenarios.

Exp	Start Position	Goal Position	Obstacle 1	Obstacle 2
Exp 1	(7.5, 7.5, 0)	(1.5, 7.5, 0.23)	(2.5, 3.5, 0.37)	-
Exp 2	(7.5, 7.5, 0)	(2.5, 3.5, 0.37)	(7, 6.5, 0.02)	(5, 5.5, 0.18)

## Data Availability

All data generated or analyzed during the study of this manuscript are included in the article.

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
