# Peer review of "Path Planning for Wheeled Mobile Robot in Partially Known Uneven Terrain"

_sensors, 2022, doi:10.3390/s22145217_

Round 1
Reviewer 1 Report
In general, it's a qualified paper with fine structures, methodologies, analyses and detailed simulation on this proposed system. Authors present a partially-unknown uneven terrain environment respecting the terrain features, which plans a global path for the robot based on the partially-unknown map, and then employs a Q-learning method for local path planning to avoid locally detected obstacles in close range as well as impassable terrain areas when the robot tracks the global path. Besides, several fine ideas are introduced in this paper. Eventually, the simulation and comparison with other schemes are also described in detail. I therefore suggest this paper can be accepted.
Author Response
We thank sincerely the editor and the reviewers for reviewing our manuscript. Thank you very much for your time and recommendation.
Reviewer 2 Report
The manuscript (sensors-1777633) intends to propose a hierarchical path planning approach for a wheeled 3 robot to move in a partially-unknown uneven terrain. Simulation and experimental studies are conducted to reveal effectiveness of the proposed method. However, the presented paper has serious issues as follows.
It should be noted that the idea of the hierarchical path planning approach is not new in literature for plan planning of mobile robots, and the combination of the A⋆ algorithm and Q-learning method is one of the common means.
In the problem description, there are too many well-known items rather than a specific characterization of the actual problem. That is, to some extent, there is no new problem here.
The same situation in the methodology as above, the reviewer cannot see any development in the method.
The analyses of results are not as deep enough, especially in the experimental part. Line 234-248, almost half of the content describes known factors in Fig. 8-9, and the other half just mentions general conclusion.
Last but least, language of the manuscript should be improved.
Author Response
Dear Reviewer 2.
We thank sincerely the editor and you for reviewing our manuscript. We have carefully revised our paper according to the comments. The revised parts have been highlighted in red in the revised manuscript.
The attached file is our detailed point-to-point response to Reviewer 2. I deeply appreciate your consideration of our manuscript, and looking forward to receiving comments from you. Thank you for your consideration.
Best regards,
Xiaoshan Bai,
Assistant Professor, College of Mechatronics and Control Engineering, Shenzhen University

Reviewer 3 Report
This paper presents path planning and obstacle avoidance for mobile robot navigation in uneven terrain. It divides the technique to global planning using A* algorithm and local planning using Q-learning. Some experimental results are presented with comparison and evaluation using simulation and small scale outdoor environment. Although the method is presented for mobile robot navigation and some demonstrations are provided, there are several issues to be clarified in the revised manuscript. First, the title of this paper is confusing. The authors use the term "partially-unknown", which actually indicates the dynamic environment change. It is not commonly used in the researchers in this field, and can cause misleading. Please make this easier to understand with general terms. Second, the authors spend a large part of content in Section 2 to model the terrain. It might be useful for better illustration, but most of the equations are so obvious and can be left off. Some important formulations directly used for calculation should be fine. Third, the algorithms in Section 3 for path planning basically are all well-known materials, including A* algorithm for global path planning and Q-learning for obstacle avoidance. When those algorithms are described, there are no clear relation for their applications to robot navigation in uneven terrain. It is highly suggested that more recent references including "Lin, H.Y. and Peng, X.Z., 2021. Autonomous Quadrotor Navigation With Vision Based Obstacle Avoidance and Path Planning. IEEE Access, 9, pp.102450-102459." and "Gia Luan, P. and Thinh, N.T., 2020. Real-time hybrid navigation system-based path planning and obstacle avoidance for mobile robots. Applied Sciences, 10(10), p.3355." should be discussed, as they provide the algorithms closer to the real-world application scenarios. Fourth, there are simulation for the work, but it is fairly limited. Since it is for synthetic data, it is expected to provide more scenarios for test. Fifth, the experiment conducted in the real-world scene does not show the value of the proposed method. The core idea is to use the 3D DEM for uneven terrain, but the real experiment with a very restricted region is difficult to demonstrate the effectiveness. In this case, local sensing might be a better way for obstacle avoidance.
Author Response
Dear Reviewer,
We thank sincerely the editor and you for reviewing our manuscript. We have carefully revised our paper according to the comments. The revised parts have been highlighted in red in the revised manuscript.
The attached file is our detailed point-to-point response to your comments. I deeply appreciate your consideration of our manuscript, and looking forward to receiving comments from you. Thank you for your consideration.
Best regards,
Xiaoshan Bai,
Assistant Professor, College of Mechatronics and Control Engineering, Shenzhen University

Round 2
Reviewer 2 Report
The reviewer has to recommend reject since most of comments have not been adequately addressed.
Author Response
Dear Reviewer,
We thank sincerely the editor and you for reviewing our manuscript. We have carefully revised our paper according to the comments. The revised parts have been highlighted in red in the revised manuscript.
The attached file is our detailed point-to-point response to your comments. Please check it, thanks.
Best regards,
Xiaoshan Bai,
Assistant Professor, College of Mechatronics and Control Engineering, Shenzhen University

Reviewer 3 Report
This revision has addressed most of the reviewer's concerns. It can be accepted after minor grammar and sentence editing.
Author Response

(The authors gave the same response as above.)
